

# Nodes with high centrality in protein interaction networks are responsible for driving signaling pathways in diabetic nephropathy

Maryam Abedi[1] and Yousof Gheisari[1,2]

[1] Department of Genetics and Molecular Biology, Isfahan University of Medical Sciences, Isfahan, Iran
[2] Regenerative Medicine Lab, Isfahan Kidney Diseases Research Center, Isfahan University of Medical Sciences, Isfahan, Iran

## ABSTRACT

In spite of huge efforts, chronic diseases remain an unresolved problem in medicine. Systems biology could assist to develop more efficient therapies through providing quantitative holistic sights to these complex disorders. In this study, we have re-analyzed a microarray dataset to identify critical signaling pathways related to diabetic nephropathy. GSE1009 dataset was downloaded from Gene Expression Omnibus database and the gene expression profile of glomeruli from diabetic nephropathy patients and those from healthy individuals were compared. The protein-protein interaction network for differentially expressed genes was constructed and enriched. In addition, topology of the network was analyzed to identify the genes with high centrality parameters and then pathway enrichment analysis was performed. We found 49 genes to be variably expressed between the two groups. The network of these genes had few interactions so it was enriched and a network with 137 nodes was constructed. Based on different parameters, 34 nodes were considered to have high centrality in this network. Pathway enrichment analysis with these central genes identified 62 inter-connected signaling pathways related to diabetic nephropathy. Interestingly, the central nodes were more informative for pathway enrichment analysis compared to all network nodes and also 49 differentially expressed genes. In conclusion, we here show that central nodes in protein interaction networks tend to be present in pathways that co-occur in a biological state. Also, this study suggests a computational method for inferring underlying mechanisms of complex disorders from raw high-throughput data.

Corresponding author
Yousof Gheisari,
ygheisari@med.mui.ac.ir

# INTRODUCTION

Chronic diseases are the leading cause of death and disability. Even with enormous investigations, the exact mechanisms of the occurrence and progression of these disorders are not yet fully discovered and in most cases the therapeutic options are not satisfying. Systems biology is a promising approach to address these limitations. Using this novel

strategy, invaluable information has been obtained on the molecular basis of various diseases such as macular degeneration, myocardial infarction, metabolic syndrome, and kidney fibrosis (*Dumas, Kinross & Nicholson, 2014*; *Ghasemi et al., 2014*; *Jin et al., 2012*; *Morrison et al., 2011*). Systems biology with its global view has an exclusive potential to extract the meaning from bulk, sometimes ambiguous data derived from *omics* technologies. It can also provide a mechanistic view via generation of mathematical models (*Azeloglu et al., 2014*; *Cedersund et al., 2008*; *Swameye et al., 2003*).

Chronic kidney disease (CKD) is a common debilitating disorder consuming a considerable fraction of health budgets (*Trivedi, 2010*). CKD secondary to diabetes mellitus, known as diabetic nephropathy, is the most common subtype. Although lots of previous studies have identified the pathogenic role of individual genes and signaling pathways in DN, a systematic holistic view has rarely been attempted for this complex disorder. Among systematics studies in DN, *Starkey et al. (2010)* have shown altered retinoic acid metabolism in diabetic mouse kidney by proteomics analysis. Similarly, using network analysis approach, *Sengupta et al. (2009)* have predicted the interaction of PTPN1 with EGFR and CAV1 in DN vascular complications. However, the functional significance of the network topology parameters has not been thoroughly assessed in this disorder.

Here, we have reanalyzed a microarray dataset originally deposited by *Baelde et al. (2004)* which compares the expression profile of glomeruli from DN and normal individuals. Their analyses revealed some differentially expressed genes (DE genes) among which VEGF and Nephrin down-regulation were confirmed by real time PCR. Also, their gene ontology (GO) analysis predicted pathways such as nucleic acid metabolism, Neuropeptide signaling pathway and Actin binding to be related to DN. Here, we reanalyzed this dataset with a different statistical significance detection method which resulted in a dissimilar number of DE genes. In addition, we have constructed a protein-protein interaction network (PPI) and employed graph theory concepts to assess the network topology. Critical nodes were then selected for pathway enrichment analysis. This computational approach may also be employed for other large datasets to deepen our understandings of chronic diseases by extracting meaningful concepts from bulk raw data of high-throughput technologies.

## MATERIALS AND METHODS

### Microarray data

The mRNA expression profile of GSE1009, deposited by *Baelde et al. (2004)*, was downloaded from the Gene Expression Omnibus (GEO) database (*Barrett et al., 2009*). In this microarray experiment, the expression of genes in the glomeruli of DN patients was compared to that of healthy individuals. For further analysis, we assessed the quality of samples by hierarchical clustering and principle component analysis (PCA) based on the data of the top DE genes. For hierarchical clustering, Euclidean distance measure and average-linkage method were applied using SUMO software (*Schwager*. http://www.oncoexpress.de/software/sumo/). PCA was performed using Multibase_2015 Excel Add-In program. The dataset was re-analyzed by GEO2R tool of GEO. In analysis by GEO2R,

three normal samples were compared to three DN samples by Student's *t*-test. For *p*-value correction, Benjamini–Hochberg false discovery rate method was applied. Genes with adjusted *p*-value of less than 0.05 were considered as differentially expressed.

### Protein-protein interaction network

Using CluePedia plugin version 1.1.3 (*Bindea, Galon & Mlecnik, 2013*) of Cytoscape software version 3.1.0 (*Shannon et al., 2003*) a PPI network was constructed for DE genes in the microarray dataset. STRING database with confidence cutoff 0.80 was used for retrieving interactions. The network topology was analyzed by Cytoscape NetworkAnalyzer tool and network topology measures such as Degree, Betweenness, Closeness Centrality, and Clustering Coefficient were calculated.

### Pathway enrichment analysis

Pathway enrichment analysis was performed using Cytoscape ClueGO plugin version 2.1.3 (*Bindea et al., 2009*). In this analysis, Bonferroni step down was applied for *p*-value adjustment and pathways with adjusted *p*-value <0.05 were selected.

## RESULTS

### The quality of the microarray dataset was assessed and differentially expressed genes were identified

In this study, we re-analyzed the microarray dataset GSE1009 which compares glomeruli samples from DN patients and healthy individuals. Comparison of the two groups by GEO2R revealed that 49 genes were differentially expressed with adjusted *p*-value <0.05 (Table S1). As different parameters such as the efficiency of RNA extraction and spot detection can influence the validity of microarray experiments, we assessed the suitability of this dataset for further analysis by unsupervised hierarchical clustering and PCA with the data of the 49 genes. Both these methods could differentiate samples based on disease state (normal or DN), indicating the acceptable quality of this dataset (Fig. 1).

### The PPI network and pathway enrichment analysis of differentially expressed genes were not informative

To investigate the interaction between the 49 selected genes from the microarray dataset, we constructed a PPI network using Cytoscape CluPedia plugin. Although various kinds of interactions with different evidences (activation, post-translational modification, binding, database, experiment) were allowed to be shown, unexpectedly, only few genes revealed to be interacting (Fig. 2A). Next, to infer pathways that are related to these 49 genes, pathway enrichment analysis was performed which showed only 5 pathways with no overlap genes. These pathways were not previously shown to be related to DN (Fig. 2B).

### Pathway enrichment analysis of central genes in the enriched PPI network could detect critical pathways in DN

Observation of the scarcity of interactions between the 49 genes that all were either up- or down-regulated in DN was unexpected. It is rational to assume that in the actual network between the genes related to DN, not all genes are regulated in the level of mRNA and
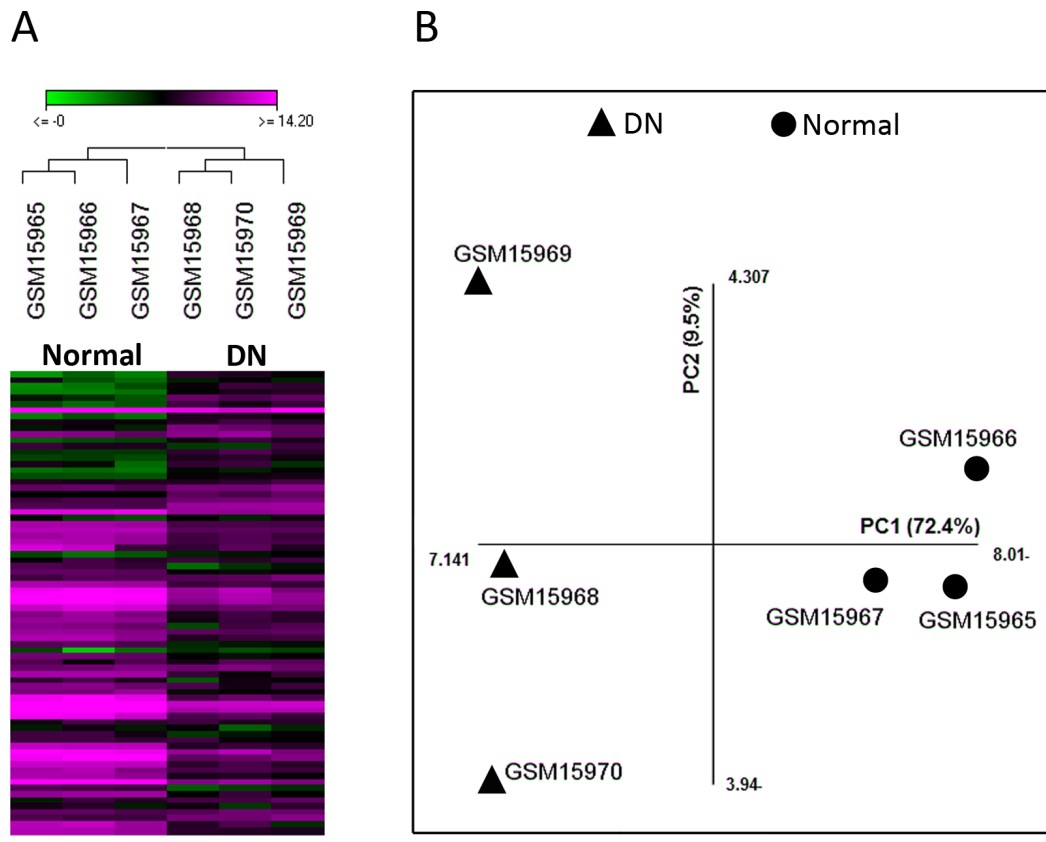

**Figure 1 The quality of microarray GSE1009 dataset is satisfying.** The heat map diagram shows the result of unsupervised hierarchical clustering for diabetic (GSM15968, GSM15969, and GSM15970) and normal (GSM15965, GSM15966, and GSM15967) samples based on the data of the top differentially expressed genes. GSM15969 is the technical replicate of GSM15968 and GSM15966 is the technical replicate of GSM15965. Each row represents a gene and each column represents a sample (A). Principal component analysis was performed on all samples based on the most up/down regulated genes. The first principle component (PC1) separates samples into DN and normal groups (B).

hence not detected in the mRNA microarray experiment. The absence of these genes makes the interaction network incomplete. Therefore, the PPI network was enriched by the addition of maximum 2 interacting nodes for each gene. This resulted in expansion of the network from 49 nodes to 137 nodes. Indeed, the added 88 genes were predicted to be interacting with the 49 initial genes based on previous knowledge. The PPI network of these 137 genes was constructed with the same parameters applied for the initial network (Fig. 3A).

Graph theory concepts such as degree, closeness centrality, and betweenness centrality were employed to assess the topology of this network. The genes were sorted based on each of these parameters and the top 15% genes with the highest rank were selected. Considering the overlapping nodes between the three gene lists, a total of 34 genes were finally chosen (Table 1). Pathway enrichment analysis was then performed starting with either the central 34 genes or the total 137 genes. Interestingly, the central gene set resulted in 62 pathways strongly related to DN (Fig. 3B). These pathways had several similar genes and formed a deeply connected network (170 edges, edge/node: 2.7). In contrast, pathway

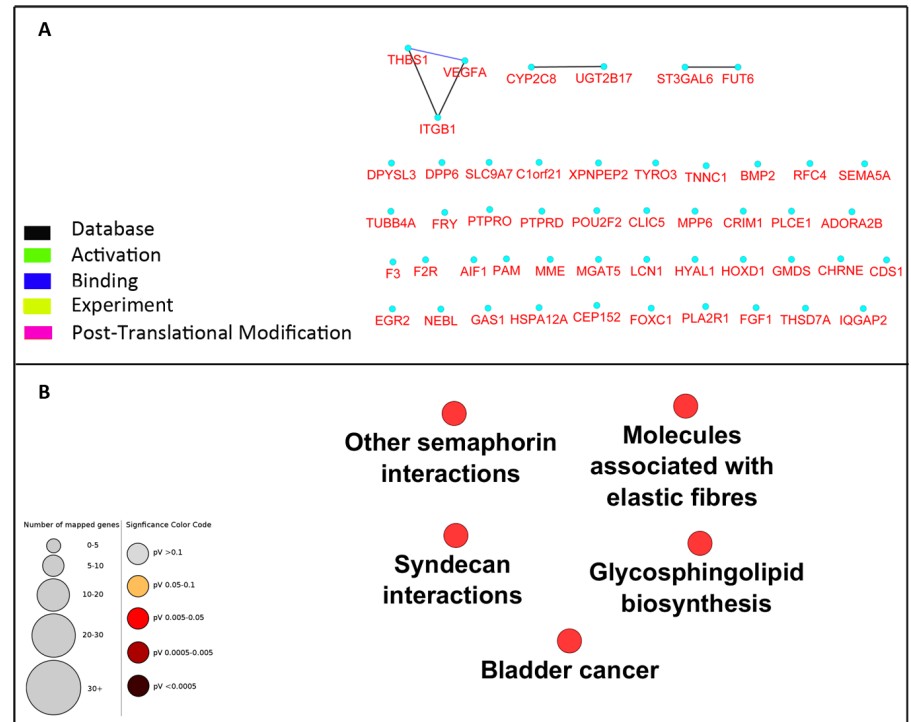

**Figure 2 Network construction and pathway enrichment analysis of differentially expressed genes were not informative.** The PPI network of the 49 differentially expressed genes has few edges as these genes do not directly interact (A). Pathway enrichment analysis of these genes could not detect critical pathways in DN. Pathways with adjusted $p$-value <0.05 are shown (B).

enrichment analysis with the total 137 genes determined 51 pathways (Fig. S1) with fewer connections to each other (86 edges, edge/node: 1.7).

## DISCUSSION

In spite of enormous studies, the current therapeutic options for most chronic diseases are not yet satisfying. It can partly be due to the simple tools and concepts of classical biology that are not appropriate for investigation of complex situations of chronic disease. The recent development of high-throughput technologies, allows the assessment of gene expression at different levels in various biological states. However, there has been a lag between the emergence of these techniques and introduction of proper mathematical methods to analyze bulk raw biological data. Therefore, for a while it was common to only inspect the few most up- or down-regulated genes individually. However, with the novel analysis methods, it is feasible to infer complex interactions at various levels from the simultaneous alteration in the expression of a bundle of genes. Therefore, re-investigation of the prior *omics* data with the current analysis tools may assist to produce valuable biomedical knowledge. In this study, the GSE1009 microarray dataset which deals with the comparison of mRNA expression profile of DN patients' glomeruli with those from healthy individuals was assessed to construct a PPI network. We found that expansion of this network followed by selection of nodes with high centrality for pathway enrichment analysis is an efficient strategy to infer critical signaling pathways in DN.

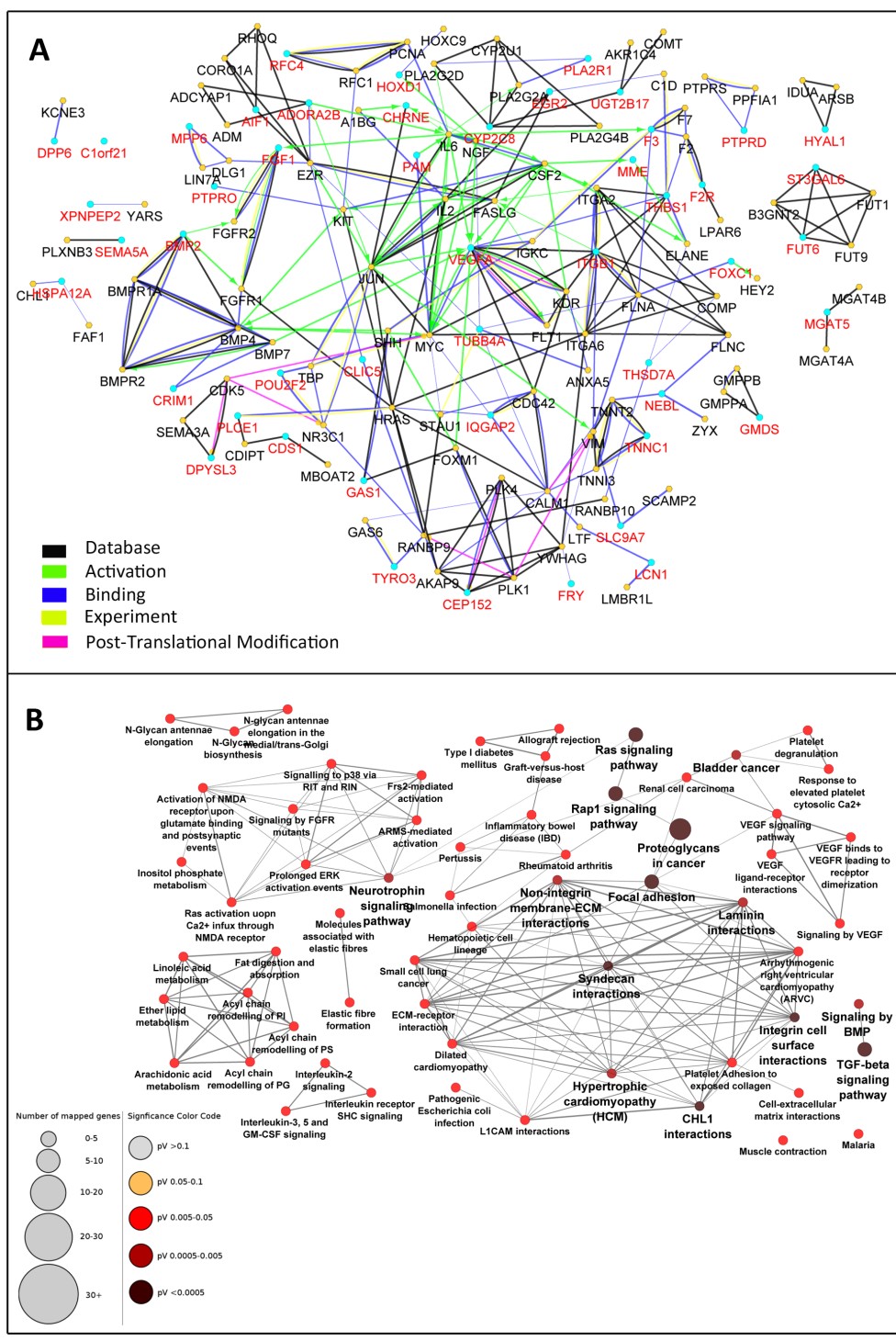

**Figure 3 Enrichment of the PPI network and selection of central nodes for pathway enrichment analysis can determine pathways essentially related to DN.** The 49-node network was extended with maximum two interactive genes for each node. The initial nodes selected from the microarray experiment are depicted with red color and enriched nodes with black (A). In this expanded network, 34 genes were selected as nodes with high centrality. Pathway enrichment analysis with these "central genes" disclosed 62 highly connected pathways related to DN. Pathways with adjusted *p*-value <0.05 are shown (B).

**Table 1  Central genes in the PPI network.** The top 15% genes with the highest degrees, betweenness centrality, and closeness centrality scores in the enriched PPI network are shown.

| Genes | Degree | Genes | Betweenness centrality | Genes | Closeness centrality |
|---|---|---|---|---|---|
| VEGFA | 23 | IL6 | 0.253 | CHL1 | 0.667 |
| JUN | 20 | JUN | 0.205 | FAF1 | 0.667 |
| ITGB1 | 17 | HRAS | 0.153 | MGAT4B | 0.667 |
| IL6 | 17 | EZR | 0.122 | MGAT4A | 0.667 |
| MYC | 17 | MYC | 0.12 | JUN | 0.408 |
| IL2 | 13 | VEGFA | 0.111 | MYC | 0.388 |
| BMP4 | 13 | VIM | 0.109 | IL6 | 0.384 |
| CSF2 | 12 | CALM1 | 0.107 | VEGFA | 0.38 |
| ITGA6 | 12 | BMP4 | 0.074 | HRAS | 0.366 |
| HRAS | 11 | ITGA6 | 0.069 | ITGA6 | 0.36 |
| KDR | 11 | CYP2C8 | 0.063 | ITGB1 | 0.359 |
| BMPR2 | 11 | PLA2G2D | 0.061 | IL2 | 0.357 |
| EZR | 11 | PLA2G2A | 0.061 | VIM | 0.354 |
| CALM1 | 11 | FLNA | 0.058 | CSF2 | 0.349 |
| FLNA | 10 | PLCE1 | 0.054 | ITGA2 | 0.341 |
| ITGA2 | 10 | TUBB4A | 0.051 | NGF | 0.337 |
| BMPR1A | 10 | THBS1 | 0.051 | TUBB4A | 0.332 |
| FGF1 | 9 | ITGB1 | 0.049 | EZR | 0.329 |
| BMP2 | 9 | IL2 | 0.049 | FLNA | 0.329 |
| FASLG | 9 | CSF2 | 0.048 | FASLG | 0.329 |
| TNNT2 | 9 | ADORA2B | 0.048 | CALM1 | 0.326 |

In this study, we found 49 genes to be differentially expressed between DN and normal samples. In contrast, in the original study; *Baelde et al. (2004)* identified 615 DE genes. This discrepancy can be due to the inappropriate bulk data analysis methods that were employed in that study. For instance, they used raw *p*-value reported by Student's *T*-test. However, it is now publicly believed that this method of statistical significance detection is associated with high false positive results. To address this problem, false discovery rate methods such as Bonferroni, Benjamini–Hochberg have been proposed for *p*-value adjustment (*Sandrine Dudoit & Callow, 2002*). Therefore, we have considered genes with adj. *p*-value <0.05 as differentially expressed.

Based on DE genes in the microarray experiment, a PPI network was constructed. Interestingly, very few interactions appeared in this network. This could be due to the fact that we had selected genes only based on mRNA expression difference and therefore, other critical genes regulated at other levels were missing. Therefore, to fill these gaps in the map of interactions, the network was expanded based on previous knowledge and a network with 137 nodes was constructed. Then we tried to determine the critical nodes in this network but as there is no simple criterion for "biologically important genes", we analyzed the topology of the network and employed a combination of different measures of centrality; some nodes such as VEGFA and JUN have high degree, so they have many

connections and are vital for the surveillance of the network. Betweenness centrality measures the number of shortest paths going through a node and so nodes with high betweenness centrality such as JUN and IL6 in this network are shortcuts of the network. In addition, nodes with highest closeness centrality such as CHL1 and FAF1 in our network are physically nearest genes to all nodes. Using these parameters, 34 genes were assumed to have high centrality.

Starting with a set of genes, pathway enrichment analysis allows determination of the top affected functions in a specific disease. An interesting finding in this study was that pathway enrichment with the set of 34 central genes was more informative than enrichment with the initial 49 genes or even with the total 137 genes in the enriched PPI network. It is widely believed that the functional significance of a protein is related to its position in the PPI network as deletion of hub proteins are more lethal compared to non-hubs, a phenomenon known as centrality-lethality rule (*Hahn & Kern, 2005*; *He & Zhang, 2006*; *Jeong et al., 2001*; *Yu et al., 2004*). Our study demonstrates that central nodes in PPI network tend to be present in pathways that co-occur in a given biological state and probably make pathway cross-links. This observation provides an explanation for the functional essentiality of the central nodes.

Pathway enrichment analysis with the central nodes had an acceptable validity as most of the enriched pathways including TGFB, VEGF, MAPK, and BMP signaling pathways were previously shown to be associated to DN in experimental studies (*Toyoda et al., 2004*; *Turk et al., 2009*; *Ziyadeh, 2008*). With this analysis, we could also determine novel pathways which their role in DN remains to be confirmed in future studies. For instance, neurotrophin signaling pathway, which has been previously shown to be related to diabetic neuropathy (*Pittenger & Vinik, 2003*), was among the enriched pathways. Similarly, we detected platelet degranulation pathway as a potential role player in DN. Previous studies have demonstrated the role of this pathway in some profibrotic disorders such as idiopathic pulmonary fibrosis (*Crooks et al., 2014*; *Wynn, 2007*).

In conclusion, we have here introduced a systems biology approach to DN as a complex biological state. Methods employed in this study may also be used for other chronic diseases to suggest novel therapies via generation of a holistic multi-level insight.

### Funding

This study was supported by Iranian council of stem cell research and technology and Isfahan University of Medical Sciences (392598). The funders had no role in study design, data collection and analysis, decision to publish, or preparation of the manuscript.

### Grant Disclosures

The following grant information was disclosed by the authors:
Iranian Council of Stem Cell Research and Technology.
Isfahan University of Medical Sciences: 392598.

## Competing Interests

The authors declare there are no competing interests.

## Author Contributions

- Maryam Abedi performed the experiments, analyzed the data, wrote the paper, prepared figures and/or tables.
- Yousof Gheisari conceived and designed the experiments, contributed reagents/materials/analysis tools, reviewed drafts of the paper.

## Supplemental Information

Supplemental information for this article can be found online at http://dx.doi.org/10.7717/peerj.1284#supplemental-information.

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
