# Peer review of "Nodes with high centrality in protein interaction networks are responsible for driving signaling pathways in diabetic nephropathy"

_PeerJ, doi:10.7717/peerj.1284_

## Round 0.1 · original submission · Major Revisions

Please be aware that we consider these revisions to be major, and your revised manuscript will probably have to be re-reviewed.

Reviewer 1 ·

Basic reporting

The authors describe a reanalysis of a microarray study of diabetic nephropathy and a network analysis of the resulting gene lists.

The introduction and discussion sections lack a description of what the original publication by Baelde et al. found and a comparison of the new results to theirs. I find it notable that they identified 96 up-regulated and 517 down-regulated genes, whereas your analysis of the very same data finds only 16 up-regulated and 33 down-regulated genes. This vast difference at the very least needs to be commented on.

Experimental design

The Materials and Methods section lacks sufficient detail to properly review the experimental design. In light of the stark difference to the results in the original publication, it would be important to point out which exact statistical test was used to identify the significantly regulated genes (i.e. not just that GEO2R was used for the purpose).

Details on how the protein-protein interaction network was constructed is particularly lacking. Telling that a certain version of the CluPedia Cytoscape plugin was used is not sufficient. This plugin is able to pull data from a variety of different resources including both physical interaction databases including both IntAct and STRING. Which source databases were used for the analysis - this is essential since IntAct is a physical interaction database, whereas STRING includes functional interactions. If STRING was used, which confidence cutoff was used when retrieving interactions?

Similarly, it is stated that network topology was analyzed using the Cytoscape NetworkAnalyzer tool. However, it is not explained which parameters were calculated, or it was assessed which of these were interesting/significant.

Validity of the findings

Due to the lack of details in the Materials and Methods section, it is impossible for me to fully assess the validity of the findings. Despite this, however, a few important issues were identified that will need to be addressed.

A very basic analysis, enrichment for GO terms among the 49 significantly regulated genes, was not performed. Doing so reveals that around half of the genes are associated with the cellular component term "extracellular exosome". This is a noteworthy result since diabetic nephropathy is known to cause major changes in the extracellular matrix.

Unfortunately, this also brings into question the entire network analysis. The problem is that no big interaction screens have been performed on extracellular proteins, which likely explains the lack of interactions among these proteins. This give a very poor starting point for any protein-protein interaction network analysis, including the ones presented in the manuscript.

Lastly, considering the vast differences in the list of significantly regulated genes identified in this reanalysis and in the original publication, some comparisons need to be performed. Are the genes identified here a subset of the original ones (i.e. more stringent)? Do the conclusions of the network analysis changes dramatically if one were to use the original gene list as a starting point? These are questions that need to be addressed.

Reviewer 2 ·

Basic reporting

The article describes a re-analysis of a diabetic nephropathy microarray dataset, originally described by Baelde et al. in 2004. The authors identified differentially expressed genes in this dataset, then constructed a protein-protein interaction network of the genes together with interacting partners. Study of the structure of this network showed that nodes with high centrality are enriched in pathways related to diabetic nephropathy.

The introduction lacks details of the previous work in this field. Diabetic nephropathy is well studied, but the large amount of work aimed at identifying the molecular mechanisms involved is not described. Similarly, the analysis of gene sets and pathways involved in disease is a very active field and the contributions of the methods developed in this article are not clear.

As this is a re-analysis of a previously published dataset, it is important to summarize the findings of the original article, and explain clearly how the differentially expressed genes identified here compare to the original analysis.

Similarly how does the approach described here compare to other published methods that take into account the structure of pathways, such as SPIA, CePa, NetGSA, among others?

Minor points include:
"Using CluPedia plugin" should be “Using CluePedia plugin”.

“Cytoscape CluGO plugin” should be “Cytoscape ClueGO plugin”.

The references are not in the correct format.

Experimental design

More details are needed to be able to evaluate the methods used in this work. For example, how are the top differentially expressed genes determined? How is the PPI network constructed? How are the interactions between proteins defined?

More details are also needed about how the pathway enrichment analysis was performed. For example, what previous knowledge was used to identify the interacting nodes for each gene?

Validity of the findings

The conclusion stated in the abstract: "this study suggests a computational method for inferring underlying mechanisms of complex disorders from raw high-throughput data", is not justified by the results of the study. The authors have not inferred any of the mechanisms underlying diabetic nephropathy.

---

## Round 0.2 · accepted · Accept

The reviewers comments and suggestions have been taken into account in the revised version of the manuscript.